# Massively parallel cantilever-free atomic force microscopy

Wenhan Cao [1], Nourin Alsharif[1], Zhongjie Huang [2], Alice E. White [1,3,4,5], YuHuang Wang [2] & Keith A. Brown [1,3,5 ✉]

Resolution and field-of-view often represent a fundamental tradeoff in microscopy. Atomic force microscopy (AFM), in which a cantilevered probe deflects under the influence of local forces as it scans across a substrate, is a key example of this tradeoff with high resolution imaging being largely limited to small areas. Despite the tremendous impact of AFM in fields including materials science, biology, and surface science, the limitation in imaging area has remained a key barrier to studying samples with intricate hierarchical structure. Here, we show that massively parallel AFM with >1000 probes is possible through the combination of a cantilever-free probe architecture and a scalable optical method for detecting probe–sample contact. Specifically, optically reflective conical probes on a comparatively compliant film are found to comprise a distributed optical lever that translates probe motion into an optical signal that provides sub-10 nm vertical precision. The scalability of this approach makes it well suited for imaging applications that require high resolution over large areas.

[1] Department of Mechanical Engineering, Boston University, Boston, MA 02215, USA. [2] Department of Chemistry and Biochemistry, University of Maryland, College Park, MD 20742, USA. [3] Division of Materials Science & Engineering, Boston University, Boston, MA 02215, USA. [4] Department of Biomedical Engineering, Boston University, Boston, MA 02215, USA. [5] Physics Department, Boston University, Boston, MA 02215, USA. ✉email: brownka@bu.edu

Since its invention in 1986, atomic force microscopy (AFM) has become the leading method for obtaining information about surface topography and functional properties at the micro- and nanoscales[1,2]. To detect the minute forces between a sharp tip and a substrate, AFM conventionally utilizes a microscopic cantilever that deflects under the influence of local forces, giving rise to a motion that can be detected using an optical lever[2,3]. However, due to the serial nature of probe-based imaging, finer spatial resolution is obtained at the cost of a smaller field-of-view[4]. Ongoing efforts to address this challenge include designing probes with higher bandwidth[5–8] and adopting arrays of probes such as the IBM Millipede[9]. However, modern imaging arrays feature only 30 probes, highlighting the difficulty in efficiently parallelizing cantilever-based sensing[10,11]. While the adoption of probe arrays by the AFM community has been limited, arrays of probes are widely used for scanning probe lithography (SPL)[12–15], or the process of defining patterns using a nanoscale physical probe through myriad means such as mechanical deformation, anodic oxidation, and direct material deposition[16–20]. To address the limited throughput inherent to serial patterning, a cantilever-free architecture has been explored, in which an array of probes rests on a compliant film on a rigid surface[21–24]. While this architecture endows the probes with the compliance needed for gentle probe–sample contact and a scalability affording up to millions of probes, the force-sensing capability afforded by the cantilever is lost. If such cantilever-free probe arrays could be modified to enable parallel detection of probe–sample contact, they could provide a means to massively parallelize AFM and transformatively increase the throughput of this impactful family of imaging tools.

Here, we demonstrate massively parallel AFM enabled by an array of probes in a cantilever-free architecture that provide local topographical information through a scalable optical mechanism that we term the distributed optical lever (Fig. 1a). By constructing a model of the distributed optical lever and systematically exploring it using coordinated force and optical microscopy, we find the optical contrast to be linear in both force and deformation and able to provide sub-10 nm vertical precision. Using probe arrays based on this architecture and imaging mechanism, we simultaneously image using 1088 probes in an array and map sample height with 100 nm lateral resolution and 9 nm vertical precision across 0.5 mm. The high-throughput nature of this system makes it promising for application in fields where both high resolution and large areas are important, such as integrated circuit metrology, optical metasurface characterization, and multi-scale studies of biological tissue.

## Results

**Cantilever-free probes as a distributed optical lever.** As a foundation for cantilever-free AFM, we postulated that vertically moving a rigid probe on a compliant backing layer will result in a visible deflection of the backing layer (Fig. 1b). Combining contact mechanics and an analytical estimate for specular reflection off a tilted surface, we developed a model of this effect, which we term a distributed optical lever (Supplementary Fig. 1 and Supplementary Information). Two key relationships emerge from this analysis. First, assuming that the force acting on the probe leads primarily to a deflection of the backing layer, contact mechanics indicates that the cantilever-free probe spring constant $k_{cf}$ is only dependent upon the probe base radius $R$ and backing layer effective elastic modulus $E$ and is found to be

$$k_{cf} = 2RE. \tag{1}$$

It should be noted that while this relationship is expected to accurately relate the deformation of the backing layer to the

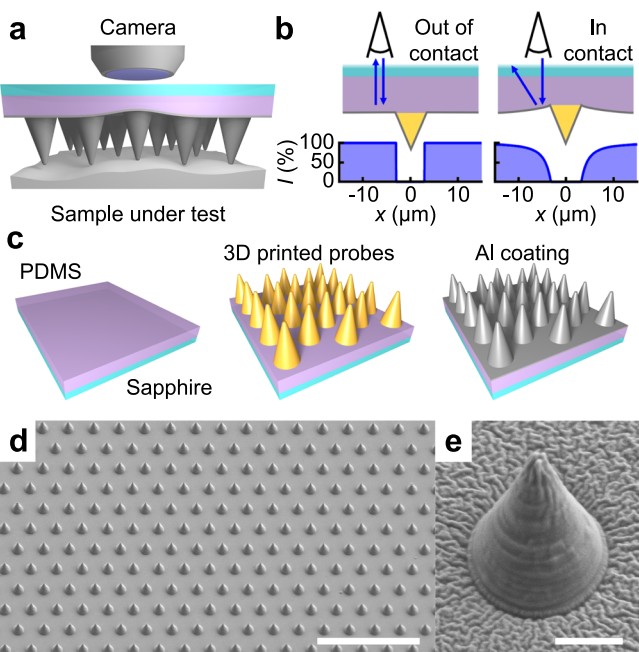

**Fig. 1 Cantilever-free atomic force microscopy. a** Scheme of the experimental setup as viewed from the side. Probe–sample contact results in deformation of the elastomer thin film. **b** Illustration of the optical contrast upon deformation of the elastomeric thin film as viewed through the sapphire wafer. The reflected light intensity $I$ changed with position $x$. The optical signature of probe motion is termed a distributed optical lever. **c** Microfabrication process used to realize probe arrays for imaging involving a sapphire wafer, polydimethylsiloxane (PDMS) backing layer, rigid polymer probes, and aluminum reflective coating. Scanning electron microscopy (SEM) images of **d** a probe array and **e** a single probe in an array with 60 μm and 3 μm scale bars, respectively.

probe–sample force, this does not preclude the probe or surface from deforming during an imaging experiment, an occurrence that is common to all forms of AFM. Such deformation effectively softens the probe–sample spring constant (i.e. the proportionality between the vertical motion of the z-piezo and the probe–sample force). For the system considered here, simulation predicts that this softening will be less than 10% provided the substrate stiffness is greater than 1 GPa (Supplementary Fig. 2). The fact that the probe is expected to behave as a linear spring is empowering as this is a foundational property of cantilever-based probes. In order to understand the optical consequences of deforming the probe, we combined an estimate of surface deflection from contact mechanics and a ray optics model of light reflected off the backing layer to quantify the change in reflected intensity $I$ as a fraction of the maximum intensity $I_{max}$, which was found to only depend on the probe motion $\delta_0$ and angular aperture $\beta$ of the optics. Specifically, at the perimeter of the probe, we compute (See Supporting Information Section 1 for derivation)

$$I = I_{max}\left(1 - \frac{2\delta_0}{\pi R \sin(\beta/2)}\right). \tag{2}$$

Interestingly, this model predicts that the change in reflected light intensity will be linear in probe motion and can provide ~1 nm precision in determining the vertical position of the probe. However, for operation without complications from the sample in contact with the probe array, the compliant backing layer must be rendered reflective. To realize rigid probes on compliant films, we used two-photon polymerization direct laser writing (2PP-DLW) to write rigid conical probes[6] directly on a backing layer composed of a polydimethylsiloxane (PDMS) film on a sapphire

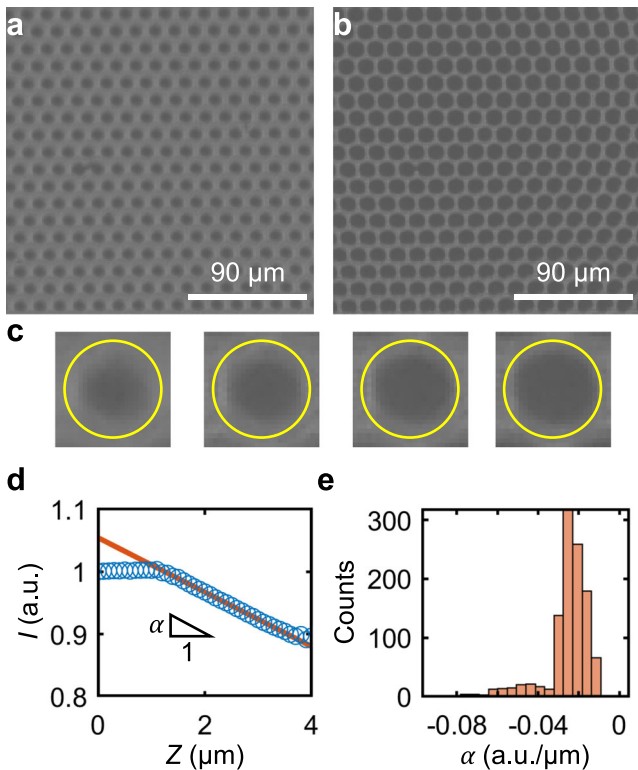

**Fig. 2 Calibrating the cantilever-free probe array.** Optical images of the cantilever-free probe array out of contact at **a** probe array extension $Z = 1$ μm and in contact at **b** $Z = 4$ μm. **c** Image of one probe with a circular region of interest (ROI) denoted by the yellow circle at $Z = 1, 2, 3,$ and 4 μm (from left to right). **d** Averaged $I$ for a single probe vs. $Z$ shown with the linear fit used to estimate slope $\alpha$. **e** Histogram of $\alpha$ from the calibration of all 1088 probes in a single array.

wafer. These structures were subsequently rendered reflective using an aluminum coating (Fig. 1c—see 'Methods' for details). In contrast with prior methods for fabricating rigid probes[25–27], 2PP-DLW enables arbitrary probe geometries with optically pristine interfaces. Inspection using scanning electron microscopy (SEM) confirmed that the final structures consisted of planar arrays of polymeric probes (Fig. 1d, e). Mechanical characterization of these probes revealed that they behaved as linear springs with ~10 N/m spring constants, which is in the range of that used in cantilever-based AFM (Supplementary Fig. 3).

In order to evaluate whether cantilever-free AFM can function in a massively parallel format, we constructed an array of 1088 probes and mounted it in a scanning probe instrument. Bright-field optical images taken through the sapphire wafer depicted an array of dark spots, each corresponding to a single probe (Fig. 2a). Subsequently, the probe array was brought in proximity with a flat silicon wafer and leveled with respect to the surface using force feedback, as is common in cantilever-free SPL[28]. To calibrate the distributed optical lever associated with each probe, the probe array extension $Z$ was increased until the force feedback registered probe array–sample contact. Notably, the size and intensity of the dark spot corresponding to each probe changed drastically upon contact (Fig. 2b). To more quantitatively analyze the change in optical contrast with increasing $Z$, image processing was used to identify the center of each probe and average the pixel brightness in a 15 μm diameter circle centered on each probe. This average pixel brightness, when normalized using an image of the probe array out of contact, was defined to be the intensity $I$ (Fig. 2c). As predicted by our model, $I$ did not change with

respect to $Z$ prior to probe–sample contact and decreased linearly with increasing $Z$ upon contact (Fig. 2d), enabling a direct translation between the optical signal and surface height. Specifically, we fit this decreasing region to $I = \alpha(h - Z)$ with slope $\alpha$ and $h$ denoting the point at which the probe makes contact with the sample—a direct measure of sample height (Fig. 2e). Critically, once probe–sample contact is made for a calibrated probe, one measurement of $I$ is sufficient to compute $h$.

**Topographical image reconstruction.** The connection between optical contrast and probe deformation indicates that it is possible to use cantilever-free AFM to generate topographical images of a surface. In particular, if the probe array is brought into contact with a surface and an optical picture is taken, this can be considered a "frame" that contains information about the height of the sample beneath each probe. Since the sample can be raster scanned in the $x$–$y$ plane with respect to the probe array, the process of bringing the probe array into contact and taking an optical image can be repeated such that subsequent frames are collected while the probe array visits a grid of points. In this way, the topographical information about the entire sample area beneath the probe array can be retrieved with a pixel density that is determined by the number of frames. It is important to highlight that the probes never move laterally while in contact with the sample. Specifically, the probes move vertically into contact with the sample, are held fixed for a contact time, and then are withdrawn until the instrument's force sensor registered that they were out of contact prior to the lateral move that brings them into their next lateral position. Because of this, abrasion that is common in contact mode imaging is not present.

To accurately reconstruct the series of optical images into a topographical image, we developed a reconstruction algorithm. First, the locations of the probes in the optical image were determined using a Hough transform (an image-processing technique that identifies circular features) and used to compute an **I** matrix based on averaging the pixel brightness around each probe (Fig. 3a). Next, **I** was converted into an **h** matrix using the probe array calibration, which was unchanging as the probe array did not move relative to the optics. This process was repeated for each frame until each probe had visited a $15 \times 15$ μm² field-of-view (Fig. 3b). Since the probes in the array were hexagonally packed with a 15 μm probe-to-probe spacing, the square fields-of-view captured by each probe overlapped with those of four neighboring probes, which facilitates stitching them into a continuous image (Fig. 3c).

Cantilever-free AFM introduces a few classes of imaging artifacts that must be corrected to accurately image a surface (Fig. 3d). For instance, the probes in the array may vary in height. However, the field-of-view overlap between neighboring probes provides an avenue for addressing this potential variability in probe height. Specifically, for a location $(x, y)$ on the sample that is visited by two probes (e.g. probes 1 and 2), we may specify the deviation in sample height $\Delta h_{1,2} = h_1(x, y) - h_2(x, y) = H_1 - H_2$ where $H_i$ is the height of probe $i$. For a probe array with $k$ total probes, there are ~2$k$ distinct overlapping regions, so this presents a linear system of equations that can be solved using a least-squares method. Thus, we compute a deviation vector $\Delta \mathbf{h}$ and a connectivity matrix **K** for defining $\Delta \mathbf{h} = \mathbf{KH}$ that can be solved as $\mathbf{H} = (\mathbf{K}^T\mathbf{K})^{-1}\mathbf{K}^T\Delta\mathbf{h}$. For this initial imaging experiment, the standard deviation of **H** was found to be 1.4 μm. In addition to variations in probe height, $Z$ can vary frame-to-frame due to the repeatability of stage motion. As a measure of this, the mean of **h** for each frame was computed and found to have a 32 nm standard deviation. This offset was removed by shifting the mean of each frame to be zero.

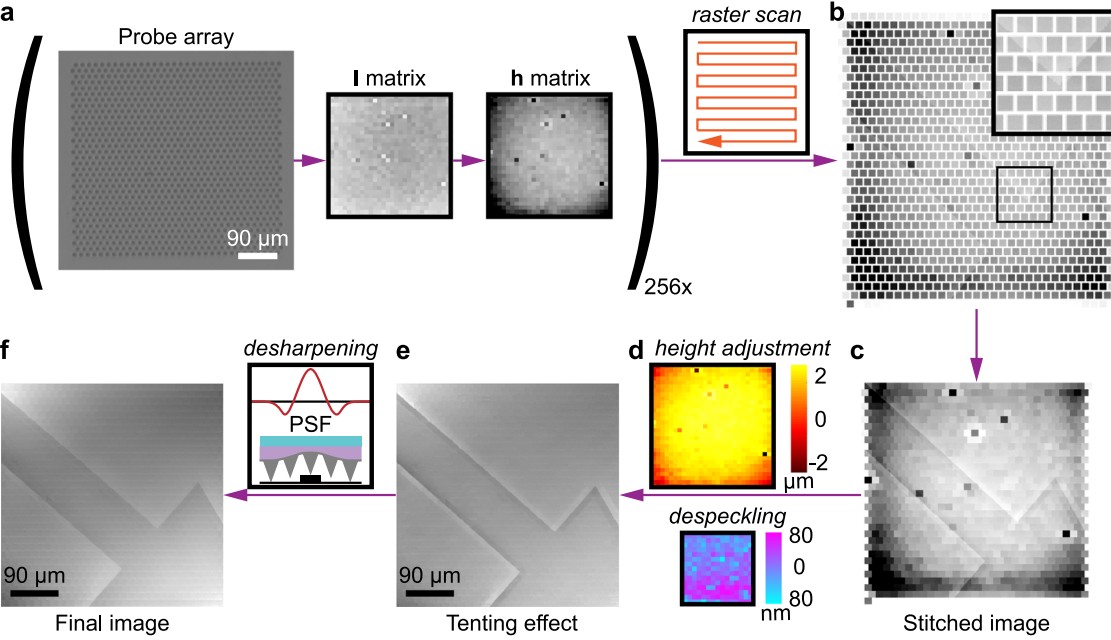

**Fig. 3 Topographic image reconstruction in cantilever-free AFM. a** An optical image taken with the probe array in contact was used to compute an intensity matrix **I**, which was used with the calibration to determine the sample height **h**. By repeating this step as the sample was raster scanned relative to the probe array, **b** the field-of-view of each probe was reconstructed to form a raw image. **c** Leveraging the overlapping regions between probes, these fields-of-view were merged to form a stitched image. **d** A probe height correction, which shifts each field-of-view by the probe height, and a despeckling process, which corrects for frame-to-frame variation in Z, were applied to correct their corresponding imaging artifacts. **e** Once reconstructed, the physical coupling between neighboring probes was evident and removed by deconvoluting the image with a point spread function (PSF) to reconstruct **f** the final image.

As a final step in the image reconstruction process, we noted that the probes were not mechanically isolated from one another due to their close proximity. Thus, deforming one probe physically moves neighboring probes, resulting in proximal probes registering an artificial change in **h**. From an image-processing perspective, this artifact is a sharpening process as it enhances the contrast between neighboring probes and is visible as a border between regions with different heights (Fig. 3e). This artifact was removed by estimating the point spread function (PSF) and using deconvolution to recover the original image. Once complete, the final reconstructed image is produced (Fig. 3f). Importantly, this empirical measure of crosstalk in which probes move vertically 35% the distance as their neighbors is in good agreement with the 29% value predicted by finite element simulations (Supplementary Fig. 4). These simulations confirm a critical feature of crosstalk, specifically that probes move a consistent fraction of the distance of their neighbors. This linearity is key as it allows imaging artifacts from crosstalk to be removed using linear image processing, thus enabling the production of an image free from crosstalk artifacts. It is worth noting that in our prior work using polymeric probes prepared using DLW, we found that continuous imaging in contact mode for 8 h did not produce a degradation in image quality[6]. Further, the intermittent contact imaging described here is known to produce less probe wear due to the lack of lateral motion during contact[29]. Thus, while tip wear remains a consideration that requires further study, it is not likely to be a major limiting feature. While we did not observe the breakage of any probes during imaging, we note that the ability of neighboring probes to overlap their imaging areas provides this approach a potential method for dealing with broken probes by increasing the imaging area such that nominally each region is imaged by multiple probes.

**Massively parallel imaging**. In order to evaluate the imaging process and image reconstruction algorithm, we ran test scans on

a number of regions of an AFM calibration sample. In particular, we began by imaging a fiducial arrow feature that had a 110 nm depth, as determined by AFM (Fig. 4a). A line scan of this sample was taken at 100 nm lateral spacing, which revealed a set of discrete vertical jumps with an average step height of 126 nm, within 15% of the AFM measurement (Fig. 4b). It is worth emphasizing that this line scan represented an aspect ratio of over 10,000 where the vertical precision is estimated as 9 nm (from the root-mean-square error in flat regions) over the >0.4 mm horizontal span. This approximated precision is in agreement with an ~6 nm estimate of the precision estimated using AFM to deform a single probe (Supplementary Fig. 5). Finally, the imaging and reconstruction process was repeated on an intricate region of the calibration sample (Fig. 4c), showing the capability of this approach to generate images of multiscale surfaces. Analysis of this image by exploring the step heights measured by four probes that measured comparable regions revealed an ~6% variation in step height measured for a single probe and an ~3% variation in the average step height measured by all four probes (Supplementary Fig. 6). These metrics begin to address the question of repeatability in cantilever-free AFM, but their modest size in this proof-of-concept study shows the potential for this technique to provide reliable measurements of nanoscale structures.

While this work provides evidence that cantilever-free AFM can provide high-resolution topographical imaging, it is important to consider the opportunities and fundamental limitations of this approach. For example, one potential benefit is cantilever-free AFM could provide more information from each frame than simply topography. For instance, initial experiments using an AFM have found that torques acting on the probe can lead to asymmetric deformation profiles that enable the measurement of lateral forces. Critically, this could allow one to measure the gradient of sample topography (Supplementary Fig. 7), in contrast to cantilever-based AFMs that are at most only sensitive

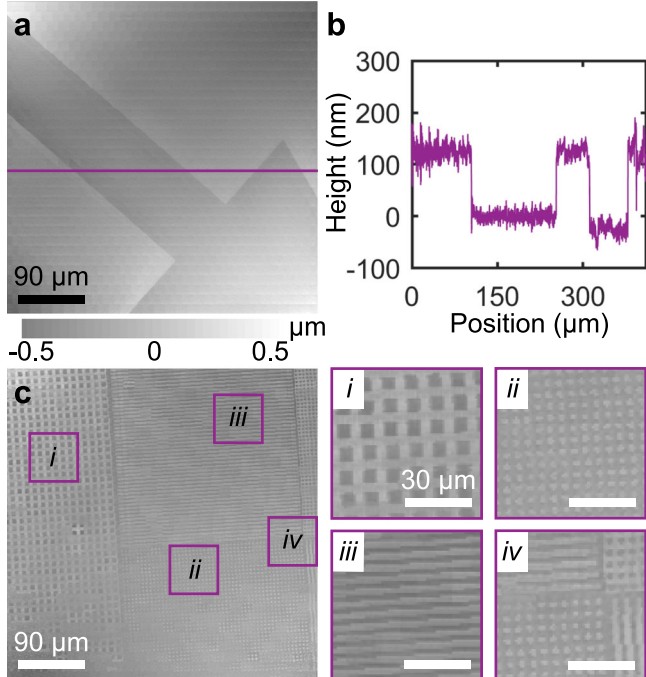

**Fig. 4 Imaging with cantilever-free scanning probes. a** Topographical image of an arrow region on a calibration grid imaged with 1 μm horizontal resolution. **b** A height profile of the arrow region taken with 100 nm horizontal resolution. Here, the vertical precision is estimated as the root-mean-square deviation of $h$ in regions expected to be flat and found to be 9 nm. **c** Cantilever-free AFM images of a complex region of the calibration sample with insets showing an assortment of lines, pits, and mesas.

to lateral forces in one direction. Further, the size of the probe array discussed here is not a fundamental limit. In particular, the maximum probe array size is limited by the optical field-of-view, which can be nearly 0.5 cm for large format sensors with 5× optical magnification. Thus, we anticipate that arrays with 100× larger area should be readily attainable. The large working area and increased throughput that are achieved by CF-AFM through parallelization come at the cost of vertical range. While the largest contrast in height that can be measured will depend upon the details of sample topography, the probe height represents an absolute maximum vertical range that can be accommodated without contact between the backing layer and the sample. This limit indicates that vertical range can be increased at the cost of lower throughput by using larger probes that are spaced further apart. Samples that deviate from planarity, such as those with substantial bowing, would be difficult to measure using CF-AFM, as all probes are subjected to the same vertical range. Taking inspiration from AFM studies that image samples multiple times with different probe tilts for accurate sidewall measurement[30], CF-AFM measurements could be repeated at multiple tilt angles to capture regions of a bowed surface.

While the experiments here were performed on stiff samples, namely silicon wafers (substrate stiffness $E_s > 100$ GPa), it is important to evaluate how this technique would perform on samples with different $E_s$. To explore this possible generalization, we performed a series of finite element simulations to compute the deformation of the backing layer-probe–sample system (Supplementary Fig. 2). In particular, high-resolution topographic imaging is possible when the sample and probe deformations are very small compared with the backing layer deformation as this will preserve a small tip–sample contact area. If, on the other hand, one wishes to deform the sample in order to study its

nanomechanical properties, a more substantial sample deformation is desired. Following these guidelines, the probes discussed herein are ideal for high-resolution topographical imaging when $E_s \gtrsim 1$ GPa and could be useful for nanomechanical studies in the range 10 MPa $< E_s < 1$ GPa. Similar to how the stiffness range of cantilever-based probes can be chosen by selecting a probe with the correct spring constant, these ranges can be shifted to higher or lower values of $E_s$ by increasing or decreasing the backing layer stiffness, respectively.

A driving consideration for parallelization is throughput; thus, imaging speed is the main concern. Here, each frame was acquired in 1 s, which indicates that the imaging bandwidth of this system (defined as the number of points in a frame divided by the time to acquire one frame) was over 1 kHz. To provide a frame of reference for this number, the imaging bandwidth of a cantilever-based system is bounded by its mechanical bandwidth, which is approximated by the cantilever resonance frequency divided by the cantilever quality factor. For standard cantilever-based probes, this can range from ~200 to 1 kHz and is a fundamental limitation of the cantilever, although it is worth highlighting achievements in the field of high-speed cantilever-based AFM that utilize high operating frequencies and custom-designed stages for fast motion[7]. In contrast, the bandwidth of cantilever-free systems can be increased through parallelization and shortening frame acquisition time. As previously discussed, scaling to arrays that are 100× larger can be achieved with no change to the optical system. Due to the simplicity of the optical measurement, we predict that the measurement scheme may be reduced to 100 ms or less without specialized optics. With these improvements, MHz bandwidth is attainable. We note that total imaging time will also be defined by the number of frames in an image and processing time for the software. For instance, the data in Fig. 4c took 256 s to collect and 54 s to process using the image-processing algorithm described in Fig. 3.

## Discussion

In summary, we have reported a massively parallel cantilever-free AFM. By designing distributed optical levers to measure the deformation of each probe, we show that scanning probes can be parallelized as a path to improving the throughput of AFM imaging. In this initial demonstration, 1088 probes were utilized in parallel to image a 5 mm wide surface with nanoscale resolution. Due to their structural simplicity and compatibility with existing lithography systems, these probe arrays could function as a stand-alone imaging tool or find use as a complement to massively parallel lithographic systems to enable simultaneous lithography and imaging in nano-combinatoric experiments[31,32]. Parallelization, while increasing the imaging bandwidth, also enforces the main limitation of this approach in the inability of probes to move independently to modulate probe–sample force or accommodate very tall features. In probe arrays such as the ones discussed here, this will impose restrictions on sample flatness and vertical range. However, it is worth highlighting work showing that cantilever-free arrays need not be passive as scanning probe lithography experiments have shown that such arrays can be independently modulated using light, heat, or pneumatic systems[13,23,33], potentially providing a path to overcoming the challenges imposed by parallelization. Interestingly, arrays of sub-wavelength apertures have been combined with the cantilever-free architecture to enable diffraction-unlimited lithography[34,35]. If the present CF-AFM method could be used in conjunction with such aperture arrays, it could potentially enable massively parallel scanning near-field optical microscopy (SNOM). This imaging approach has thus opened the door for rapid and high-resolution interrogation of surface topography for diverse

applications ranging from tissue engineering to inspection of optical metasurfaces and integrated circuits.

## Methods

**Fabrication of cantilever-free probe arrays.** Substrates for probe arrays were prepared by spin-coating a sapphire wafer with polydimethylsiloxane (PDMS) (Sylgard 184 prepared with a 25:1 base:crosslinker ratio by weight) at 1000 RPM for 60 s and then curing it at 100 °C for 1 h. The high base:crosslinker ratio was selected to render the film compliant and minimize the probe spring constant. Subsequently, DLW was used to print probes on the elastomer surface (Photonic Professional GT—Nanoscribe). Probes were written using IP-dip resin procured from Nanoscribe at 63× magnification with slicing and hatching distances of 100 nm to ensure solid and smooth structures. In addition to conical probes, cylindrical probes were also printed for mechanical characterization (Supplementary Information). To render the probes reflective, the surface of the probe array was coated with a 30 nm layer of Al (99.99% Aluminum pellets—Kurt J. Lesker Company) using an electron beam evaporation (EvoVac—Angstrom Engineering) platform at 0.2 Å/s deposition rate.

**Cantilever-free imaging.** In a typical imaging experiment, a cantilever-free probe array was mounted in a scanning probe instrument (Tera-print TERA-Fab E series) with the probes pointed downward. This system allowed the probe array to move vertically and tilt under piezoelectric control while the stage beneath the probe was moved in the $x$–$y$ plane under piezoelectric control. To this system, we mounted a high-resolution camera (Point Gray Grasshopper GS3-US-32S4C-C) and a 10× Mitutoyo objective lens (NA = 0.28). An AFM calibration sample (MikroMasch TGXYZ02) was placed on the sample stage to serve as the sample to be imaged. To obtain an image, the sample was raster scanned in a square pattern $15 \times 15\,\mu m^2$ with a step size of 1 μm. The scan size was determined by the probe array pitch as the probes were printed on a hexagonal grid with a 15 μm spacing. In addition, high-resolution line scans were also performed with a 100 nm step size.

**Image reconstruction.** For each sample imaged, a video in AVI format was recorded of the entire raster scan process. This video was imported into MATLAB and the locations of the 1088 probes were found using a Hough transform. The Hough transform is a linear transformation that is applied to a matrix to locate circular features. Here, the process entails specifying a radius of interest in units of pixels. Then, a transformed image is produced in which the value of each pixel is determined by the intensities of the pixels in the original image on a circle with the specified radius centered on the pixel's position.

These coordinates remained the same throughout the scan. Next, an intensity matrix **I** was obtained by averaging the pixel brightness in a 15 μm diameter circle centered on each probe. Subsequently, **I** was converted into a height matrix **h** using the probe array calibration. This process was repeated for each frame so that fields-of-view for each individual probe were restored in 1088 square matrices. Given the hexagonal packing of the probe array, the square fields-of-view captured by each probe overlapped with those of four other probes, which facilitates stitching them into a large continuous image matrix. A deviation vector **Δh** was computed as the difference in **h** registered for each pen in each overlapping region. This is paired with a connectivity matrix **K** for defining $\Delta h = KH$, where **H** is the height of each probe. Further, a consistent speckle pattern present in each probe's field-of-view was removed by reducing each **h** by the average **h** taken by all probes in that frame. Finally, the physical crosstalk between neighboring was removed by deconvoluting the image with a PSF that assumes neighboring pens move 35% as much as a deformed pen.

## Data availability
The data that support the findings of this study are available from the corresponding author upon reasonable request.

## Code availability
This custom MATLAB code is available at kablab.org/data.

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

## Acknowledgements

This work was supported by the Air Force Office of Scientific Research (Multidisciplinary University Research Initiatives FA9550-16-1-0150) and the National Science Foundation (NSF-1661412). N.A. acknowledges support from a BUnano Cross-Disciplinary Fellowship. The authors acknowledge support from the Boston University Photonics Center.

## Author contributions

W.C. and K.A.B. conceived and designed all experiments. A.E.W contributed to the development of the process for fabricating cantilever-free probe arrays and Y.W. aided in the development of the imaging process. W.C. and N.A. fabricated cantilever-free probe arrays. W.C. and Z.H. conducted imaging experiments. W.C. and K.A.B. developed the image reconstruction process. All authors wrote and approved the manuscript.

## Competing interests

K.A.B. has a financial stake in TERA-print LLC. The remaining authors declare no competing interests.
