## [Peer Review File · Nature Communications]

Reviewers' Comments:

Reviewer #1:

Remarks to the Author:

This is a very interesting concept for scale up of AFM imaging to large arrays. There are a few topics I would like to see addressed in more depth before paper is ready for publication.

The analysis of the tip stiffness is interesting and useful. The main body of the manuscript treats this at a high level and the interesting technical details are in the supplement. Please pull some analytical and quantitative details into the main body of the manuscript.

The tip body is relatively stiff with a spring constant in the range 10-30 N/m. At this high spring constant, I suspect that there is substantial tip deformation and wear. Please analyze and discuss the deformation of the tip in contact with a hard surface (silicon, silica) and also in contact with a soft surface (polymer, tissue). I would like to see a plot of tip deformation vs displacement for these two types of surfaces. It would also be useful to see contact pressure as a function of displacement or loading force. The manuscript should discuss how tip deformation affects the resolution.

I would like to see images of tip wear after a reasonable amount of imaging (let's say 2 hours) on a hard surface. These images should accompany a discussion of the relevant technical issues.

A limitation of the proposed approach is that it will be difficult to measure tall features. The tip becomes very stiff for large deformations. Please analyze and discuss the limits of measuring tall features and how these limits could be overcome with engineering modifications.

The manuscript mentions cross talk between neighboring tips without showing a detailed analysis. Certainly the cross talk is a source of error, and the error might be large when there are sharp changes in topography. Please show modeling, calculations, and measurements that explain the cross talk, its impact on measurement error, and a discussion of how this error limits measurements on tall features.

Reviewer #2:

Remarks to the Author:

This manuscript reports a very interesting and novel development of cantilever-free parallel force (topography) imaging system using polymer pen arrays coupled with an optical imaging, simulation, and algorithm system. They show, with this setup, the ability to massively image over a relatively large area and a good level of accuracy and resolution when compared with single probe AFM. Overall, the paper is well written. I can imagine the many more development for imaging in the future with it. I would suggest some additional data and discussions of the technology as follows.

1. at the end of the paper, the authors demonstrate massive parallel imaging with different features. How about imaging features of different heights over a large area? This would be more challenging I think. The authors should provide data on this as it would be another critical aspect in the parallel imaging, apart from the feature size.
2. how is the repeatability over multiple scans and scans over different tips of one object?
3. is there any issue on the lateral abrasion of the tip during scan?
4. the author should discuss and compare with technology developed by Mirkin and Huo et al on using PPL for optical imaging and massive cantilever-free equivalent to SNOM.

5. imaging over a large area faces issues of blowing of essentially any plane, how does this technology address (if it can) or how would be limit of detection because of this ?

Reviewer #3:

Remarks to the Author:

The paper from Cao et al. describes a massively parallel AFM measurement, comprising an array of tips mounted on a soft PDMS layer, supported by a transparent sapphire substrate.

Displacement of a tip from its unloaded position is optically visible as a change in light intensity when viewed through the sapphire. A camera, together with image analysis tools, can therefore be used to deduce the displacement of each tip in the array in parallel, potentially leading to a massive increase in AFM imaging speed.

The paper is well written and well studied. I appreciate that obviously a lot of work has gone in to the image analysis. I especially liked the fact that the raster scanning was designed so that some areas of the sample was imaged with multiple tips, allowing for corrections due to difference in tip height. Also, the potential ability to detect the slope of the surface seems very interesting.

For these reasons I recommend accepting the paper for publication, provided the authors address the following:

1)

The main goal of this technique is improved imaging speed, but the authors never show what speed they have reached, or what speed could potentially be reached. What is for instance the exposure time for one a single frame? What is the time for the entire raster, and what was the data processing time? How does these pixel rates and full image times compare to regular AFM?

2)

I think the authors should be a bit more admitting to the drawbacks of this measurement compared to regular AFM. The two main issues I would like mentioned / discussed is the lack of force control for the individual tips and the increased risk of tip damage. Regarding the first point, if I understand it correctly a 100 nm high feature in the sample will necessarily be imaged with 1000 nN more force on its highest part compared to the lowest. What implications does this have on the usefulness of the measurement? Regarding the second point, given that there are 1000 tips instead of one, the risk of at least one breaking must be very high. How stable were the polymer tips? Did you see any degradation during imaging?

3)

Minor point: I was not aware of the Hough transform prior to reading the paper, and I think many other potential readers will be in the same position. A brief description of how it works and what it actually does would be useful. Placed either where it is first mentioned (page 4) or in the Methods part.

REVIEWER COMMENTS

Reviewer #1:

This is a very interesting concept for scale up of AFM imaging to large arrays. There are a few topics I would like to see addressed in more depth before paper is ready for publication.

We appreciate the words of support and have numbered the subsequent comments for ease of organization.

1. The analysis of the tip stiffness is interesting and useful. The main body of the manuscript treats this at a high level and the interesting technical details are in the supplement. Please pull some analytical and quantitative details into the main body of the manuscript.

We thank the Reviewer for making this suggestion and we agree that the main manuscript text would benefit from the theoretical framework presented in the supplementary information (SI). While it is not practical to include the entirety of the theory, we have reproduced Equations 1 and 10 from the SI into the main text. These equations justify two key relationships that enable imaging with cantilever-free probes, namely that force is proportional to deflection (*i.e.* the probes behave as simple springs) and the derived result that the optical contrast will also be linear with deflection. We have included the following passage into the results section presenting and discussing these theoretical results:

Two key relationships emerge from this analysis. First, assuming that the force acting on the probe leads primarily to a deflection of the backing layer, contact mechanics indicates that the cantilever-free probe spring constant k_{cf} is only dependent upon the probe base radius R and backing layer effective elastic modulus E and is found to be,

$$k_{cf} = 2RE. \quad (1)$$

The fact that the probe is expected to behave as a linear spring is empowering as this is a foundational property of cantilever-based probes. In order to understand the optical consequences of deforming the probe, we combined an estimate of surface deflection from contact mechanics and a ray optics model of light reflected off the backing layer to quantify the change in reflected intensity I as a fraction of maximum intensity I_{max} , which was found to only depend on the probe motion δ_0 and angular aperture β of the optics. Specifically, at the perimeter of the probe, we compute (See Supporting Information Section 1 for derivation),

$$I = I_{max} \left(1 - \frac{2\delta_0}{\pi R \sin(\beta/2)} \right). \quad (2)$$

2. The tip body is relatively stiff with a spring constant in the range 10-30 N/m. At this high spring constant, I suspect that there is substantial tip deformation and wear. Please analyze and discuss the deformation of the tip in contact with a hard surface (silicon, silica) and also in contact with a soft surface (polymer, tissue). I would like to see a plot of tip deformation vs displacement for these two types of surfaces. It would

also be useful to see contact pressure as a function of displacement or loading force. The manuscript should discuss how tip deformation affects the resolution.

We appreciate the Reviewer raising the important points of tip deformation and wear. The issue of wear is raised more specifically in point (3), so we will focus our discussion of wear in our response to that comment. The cantilever-free probe spring constant k_{cf} being 10-30 N/m reflects the deformation of the backing layer. Whether or not the conical probe tip deforms depends upon how the tip-sample spring constant compares with k_{cf} . As the conical probe is >1000 times stiffer than the elastomeric backing layer, there is reason to be optimistic that the backing layer will absorb the bulk of the deformation. In order to explore this more fully, we performed a series of finite element simulations to determine probe deformation as suggested by the Reviewer. This analysis, as well as an image of the deformation and a table of the results, has been added to the SI as a new Section 5. Specifically, the added section reads:

5. Estimation of Probe and Sample Deformation

As the conical probe itself is composed of a polymeric material, it is conceivable that tip deformation could limit resolution or otherwise interfere with the ability to quantitatively map sample topography. In analogy with cantilever-based AFM measurements, one can define the vertical z-piezo motion as ΔZ and the deflection of the backing layer (or in the case of cantilever-based AFM, the deflection of the cantilever) as δ_0 . In addition to backing layer deformation, the probe tip can deform with magnitude δ_p in a manner that would reduce the total probe height h_p . Further, the sample itself can experience a deformation – or indentation – which can be defined as δ_s . These motions are interrelated by the expression,

$$\Delta Z = \delta_0 + \delta_s + \delta_p. \quad (14)$$

In conventional cantilever-based AFM, $\delta_p \ll \delta_0$ as cantilevers are generally chosen to be much softer than the tip-sample spring constant. Further, for typical imaging applications, probe stiffness is chosen such that $\delta_s \ll \delta_0$, which simplifies calibration of the probe and facilitates quantitative topographical mapping. In contrast, nanomechanics measurements are often performed with $\delta_s \sim \delta_0$. Such sample deformation is a critical part of quantitatively evaluating the mechanics of the underlying sample.

In order to determine the importance of each of these deflections for cantilever-free AFM, we performed a series of finite element simulations. Specifically, we performed an axisymmetric simulation using COMSOL designed to match as closely as possible the experimental conditions. A 3 GPa conical probe with tip radius $\rho = 100$ nm, 6 μm height, and 3 μm bottom radius was positioned on a backing layer with 300 kPa modulus. A sample with modulus E_s was positioned in contact with the probe and then moved ΔZ towards the probe (Figure S5a). This process allowed us to compute δ_s , δ_p , and δ_0 for a given ΔZ and E_s . Exploring $\Delta Z = 100$ nm as a typical indentation, the deformation of the sample follows an expected distribution based upon E_s (Figure S5b and Table S1). Specifically, at $E_s > 1$ GPa, nearly all of the deformation is localized in the backing layer while when $E_s < 10$ MPa, the backing layer deforms very little and the sample is indented to a great extent. This leads us to conclude that, for this probe geometry, topographic imaging is possible

when $E_s > 10$ MPa as this region has a substantial backing layer deformation. Nanomechanical measurements require appreciable sample deformation and, thus, nanomechanical experiments should be possible when $1 \text{ GPa} > E_s > 10 \text{ MPa}$.

It is also important to consider how the contact area will change upon contact. These simulations provide an avenue to calculate contact area using the approximation that, in Hertzian contact between a spherical tip and a planar substrate, the contact radius a is given by,

$$a = \sqrt{\rho(\delta_p + \delta_s)}. \quad (15)$$

This reflects that, while the maximum δ_p observed for this range is $\sim 4\%$ of ΔZ , the tip-sample contact area can also be increased through the deformation of the sample. Interestingly, the smallest a is observed at high E_s and low ΔZ , showing that these conditions would be ideal for high resolution topographical imaging (Figure S5c).

Table S1. Finite element simulation results of mechanics of different sample modulus at z-piezo motion $\Delta Z = 100$ nm for a variety of sample moduli.

Sample modulus E_s	Probe height decrease δ_p (nm)	Backing layer deflection δ_o (nm)	Sample deflection δ_s (nm)
300 kPa	0.01	0.67	99.33
1 MPa	0.02	2.19	97.79
3 MPa	0.06	6.28	93.65
10 MPa	0.18	18.5	81.36
30 MPa	0.44	41.04	58.52
100 MPa	1.00	68.66	30.35
300 MPa	1.57	83.97	14.46
1 GPa	2.48	91.66	5.86
10 GPa	3.89	95.42	0.69
100 GPa	4.17	95.76	0.07

Figure S5. (a) Finite element simulation of the deformation δ associated with probe-sample contact of a cantilever-free probe and sample with $E_s = 1$ GPa. (b) Simulated substrate indentation δ_s , backing layer deformation δ_o , and probe compression δ_p for a cantilever-free probe indenting a substrate with stiffness E_s . The simulation is performed by moving the sample $\Delta Z = 100$ nm. (c) Calculated contact radius a as a function of E_s at different ΔZ .

To emphasize the key results of this analysis in the main text, and provide context for how to improve upon the limitations of this approach, we have included a new paragraph in the discussion:

While the experiments here were performed on stiff samples, namely silicon wafers (substrate stiffness $E_s > 100$ GPa), it is important to evaluate how this technique would perform on samples with different E_s . To explore this possible generalization, we performed a series of finite element simulations to compute the deformation of the backing layer-probe-sample system. In particular, high resolution topographic imaging is possible when the sample and probe deformations are very small compared with the backing layer deformation as this will preserve a small tip-sample contact area. If, on the other hand, one wishes to deform the sample in order to study its nanomechanical properties, a more substantial sample deformation is desired. Following these guidelines, the probes discussed

herein are ideal for high resolution topographical imaging when $E_s \gtrsim 1$ GPa and could be useful for nanomechanical studies in the range $10 \text{ MPa} < E_s < 1 \text{ GPa}$. Similar to how the stiffness range of cantilever-based probes can be chosen by selecting a probe with the correct spring constant, these ranges can be shifted to higher or lower values of E_s by increasing or decreasing the backing layer stiffness, respectively.

3. I would like to see images of tip wear after a reasonable amount of imaging (let's say 2 hours) on a hard surface. These images should accompany a discussion of the relevant technical issues.

The process of using direct laser writing to realize probes is based upon our prior work using this approach to write cantilever-based probes for direct AFM imaging (N. Alsharif, A. Burkatovsky, C. Lissandrello, K. M. Jones, A. E. White and K. A. Brown, *Small* 14 (19), 1800162 (2018) or Ref 6 in the manuscript). In this prior work, we found that scanning continuously in contact mode for eight hours did not produce a degradation in image quality (Figure 3 of this reference). In contrast with contact-mode scanning in which the tip of the probe is effectively being polished off during scanning, intermittent contact as described in the present work features no lateral motion while coming in and out of contact with the sample and thus is superior from a wear perspective (see newly added Ref 29). Based upon these results, we do not anticipate wear presenting a major roadblock. We have added a passage to this effect in the main text,

It is worth noting that in our prior work using polymeric probes prepared using DLW, we found that continuous imaging in contact mode for eight hours did not produce a degradation in image quality⁶. Further, the intermittent contact imaging described here is known to produce less probe wear due to the lack of lateral motion during contact²⁹. Thus, while tip wear remains a consideration that requires further study, it is not likely to be a major limiting feature.

4. A limitation of the proposed approach is that it will be difficult to measure tall features. The tip becomes very stiff for large deformations. Please analyze and discuss the limits of measuring tall features and how these limits could be overcome with engineering modifications.

We appreciate the Reviewer highlighting this important issue. We agree that it is important to discuss the vertical range of this approach. One clarification to make, however, is that the probe does not become appreciably stiffer with higher deformation. As shown by the contact mechanics model, the finite element simulations in the new SI Section 5, and the experiments shown in Figure S4, the cantilever-free probe behaves as a linear spring within the investigated range. Thus, a more practical and absolute limit to the vertical range will be the probe height. In this work, we chose the probe height to be $6 \mu\text{m}$ but larger heights could be chosen with the caveat that larger probe spacings would likely be required to maintain adequate mechanical support and isolation between probes. Thus, a tradeoff emerges between vertical range and imaging throughput. It should be noted that conventional AFM systems can accommodate structures with a $\sim 10 \mu\text{m}$ vertical range, so the expected several μm range of these probes is a competitive proof-of-principle. We have added a passage to the text to highlight these important issues:

The large working area and increased throughput that is achieved by CF-AFM through parallelization comes at the cost of vertical range. While the largest contrast in height that

can be measured will depend upon details of sample topography, the probe height represents an absolute maximum vertical range that can be accommodated without contact between the backing layer and the sample. This limit indicates that vertical range can be increased at the cost of lower throughput by using larger probes that are spaced further apart.

5. The manuscript mentions cross talk between neighboring tips without showing a detailed analysis. Certainly the cross talk is a source of error, and the error might be large when there are sharp changes in topography. Please show modeling, calculations, and measurements that explain the cross talk, its impact on measurement error, and a discussion of how this error limits measurements on tall features.

We thank the Reviewer for the suggestion and have included additional simulations and discussion to highlight the role of crosstalk in this type of measurement. Specifically, we have included a three dimensional finite element simulation of an array of seven probes to elucidate the effect that moving one probe has on its neighbors. This is included in the SI as a new section:

6. Estimation of Crosstalk

The deformation of one probe may lead to the motion of neighboring probes by virtue of the mechanical coupling through the common backing layer. This effect will lead to imaging artifacts as the sample height registered by a probe will be influenced by the height registered by neighboring probes. In order to explore this effect, we performed a three dimensional finite element simulation (COMSOL) in which a central probe was modeled as a cylinder such that a displacement boundary condition can be applied without any complications associated with internal deformation of the probe. Six conical probes were arranged in a hexagonal array around the central probe. The base radii of the conical probes and the radius of the central cylinder were all set to 3 μm and the probe-to-probe distance was set to 15 μm to match experimental values. Once set up, a 1 μm downward displacement was applied to the central cylinder and the resulting displacement of the system was computed (Figure S6). Importantly, the neighboring probes were observed to move 0.29 μm , providing an estimate for crosstalk. It is also clear from the calculations that increasing probe-to-probe spacing will decrease crosstalk with near negligible crosstalk at $\sim 30 \mu\text{m}$. Thus, while crosstalk can be compensated for using post processing of images, it may also be removed at the cost of reduced sample throughput.

Figure S6. Three dimensional finite element simulation of crosstalk by calculating δ of a seven probe array in which the central probe (modeled as a cylinder to reduce the impact of internal deformation) is deformed by 1 μm .

To introduce these results in the main text, we have included a passage in the results and discussion section where we discuss the process to remove crosstalk.

Importantly, this empirical measure of crosstalk in which probes move vertically 35% the distance as their neighbors is in good agreement with the 29% value predicted by finite element simulations (Figure S6).

The influence of a common backing layer on the ability of this approach to image tall features is captured by the limitations in vertical range inherent to this technique, which we have addressed in Comment (4) by Reviewer 1.

Reviewer #2:

This manuscript reports a very interesting and novel development of cantilever-free parallel force (topography) imaging system using polymer pen arrays coupled with an optical imaging, simulation, and algorithm system. They show, with this setup, the ability to massively image over a relatively large area and a good level of accuracy and resolution when compared with single probe AFM. Overall, the paper is well written. I can imagine the many more development for imaging in the future with it. I would suggest some additional data and discussions of the technology as follows.

1. at the end of the paper, the authors demonstrate massive parallel imaging with different features. How about imaging features of different heights over a large area? This would be more challenging I think. The

authors should provide data on this as it would be another critical aspect in the parallel imaging, apart from the feature size.

We agree that ability to measure features of different heights – or vertical range – is a critical issue. As we addressed this in our response to Comment 4 by Reviewer 1, we direct the Reviewer to this passage. That being said, we emphasize that, within the vertical range allowed by the instrument, the principles described herein lead us to believe that it will not be more challenging to image structures with larger variations in height. This is because of the linear relationships between motion, force, and optical contrast.

2. how is the repeatability over multiple scans and scans over different tips of one object?

We agree that the issue of repeatability is an important consideration to raise. As noted by the Reviewer, this could relate to imaging repeatability by a single probe or variations between probes. Thus, to discuss both types, we have included additional analysis and discussion in the manuscript and SI. In particular, we have added the following section to the SI:

7. Quantification of Repeatability

The ability of a measurement system to repeatably measure features is an important metric to consider. In an array-based system, this has two distinct facets. Specifically, one could describe the ability of a single probe to repeatably measure a feature (single probe repeatability) or one could describe the ability of multiple probes to measure the same feature (repeatability between probes). Measurements of known calibration samples provide a measure of these metrics. Specifically, the image taken of the fiducial arrow provides a useful dataset to explore these as we may identify a number of probes that image ostensibly identical regions of the arrow feature and compare their images (Figure S7). In choosing four probes, (ID = 430, 562, 618 and 684), we find the step heights to be 119.8 ± 7.2 nm, 110.7 ± 6.6 nm, 113.1 ± 7.3 nm, and 115.0 ± 6.9 nm. These data can be analyzed to evaluate the single probe repeatability as the average coefficient of variation of these measurements, which was found to be ~6%, and the repeatability between probes as the coefficient of variation of the average found by each probe, which was found to be ~3%.

Figure S7. Cantilever-free atomic force microscope image with zoomed in regions and line cuts corresponding to four probes (ID = 430, 562, 618 and 684).

We have also included the results of this analysis in the main text through the passage:

Analysis of this image by exploring the step heights measured by four probes that measured comparable regions revealed a $\sim 6\%$ variation in step height measured for a single probe and a $\sim 3\%$ variation in the average step height measured by all four probes. These metrics begin to address the question of repeatability in cantilever-free AFM, but their modest size in this proof-of-concept study shows the potential for this technique to provide reliable measurements of nanoscale structures.

3. is there any issue on the lateral abrasion of the tip during scan?

We thank the Reviewer for raising this point as a lack of lateral motion is a major advantage of this intermittent contact approach. Here, the probes are lowered directly vertically into contact and held fixed for a specified time, then retracted out of contact prior to lateral motion. Because of this, probes never experience any abrasion that is common in contact mode imaging. We have added a passage to highlight this,

It is important to highlight that the probes never move laterally while in contact with the sample. Specifically, the probes moved vertically into contact with the sample, are held fixed for a contact time, and then are withdrawn until the instrument's force sensor registered that they were out of contact prior to the lateral move that brings them into their next lateral position. Because of this, abrasion that is common in contact mode imaging is not present.

4. the author should discuss and compare with technology developed by Mirkin and Huo et al on using PPL for optical imaging and massive cantilever-free equivalent to SNOM.

The work referenced by the Reviewer (namely F. Huo, G. Zheng, X. Liao, L. R. Giam, J. Chai, X. Chen, W. Shim and C. A. Mirkin, Nature Nanotech. 5, 637-640 (2010)) describes the use of a similar architecture for near field optical lithography, but does not demonstrate microscopy. That is not to say that microscopy would be impossible, but it appears not to have been demonstrated at this point. We have added a passage to the conclusion highlighting this possibility and how the present work could enable such future progress,

Interestingly, arrays of sub-wavelength apertures have been combined with the cantilever-free architecture to enable diffraction-unlimited lithography^{34,35}. If the present CF-AFM method could be used in conjunction with such aperture arrays, it could potentially enable massively parallel scanning near-field optical microscopy (SNOM).

5. imaging over a large area faces issues of bowing of essentially any plane, how does this technology address (if it can) or how would be limit of detection because of this ?

We agree that planarity is a requirement for this approach, at least in the instantiation described herein. The degree to which a surface can deviate from a plane is expected to be commensurate with the micron-scale vertical range described above. We have added text to the conclusions acknowledging this limitation and opportunity for future study,

Samples that deviate from planarity, such as those with substantial bowing, would be difficult to measure using CF-AFM as all probes are subjected to the same vertical range. Taking inspiration from AFM studies that image samples multiple times with different probe tilts for accurate sidewall measurement³⁰, CF-AFM measurements could be repeated at multiple tilt angles to capture regions of a bowed surface.

Another imaging consideration that this raises is bowing artifacts where an image is subject to a gradual curvature across the entire region. This generally occurs because moving the piezo stage can slightly change the probe-sample separation. We note that while all probe-based imaging tools are susceptible to such artifacts, CF-AFM actually has comparative advantages as each probe is individually calibrated to find contact. Further, the distance moved by the piezo is comparatively small (here $\sim 15 \mu\text{m}$), so variations in contact will be minimized. Finally, such bowing artifacts, if present, would be apparent in the data captured by each probe and therefore could be robustly removed in a single correction operation.

Reviewer #3:

The paper from Cao et al. describes a massively parallel AFM measurement, comprising an array of tips mounted on a soft PDMS layer, supported by a transparent sapphire substrate. Displacement of a tip from its unloaded position is optically visible as a change in light intensity when viewed through the sapphire. A camera, together with image analysis tools, can therefore be used to deduce the displacement of each tip in the array in parallel, potentially leading to a massive increase in AFM imaging speed.

The paper is well written and well studied. I appreciate that obviously a lot of work has gone in to the image analysis. I especially liked the fact that the raster scanning was designed so that some areas of the sample was imaged with multiple tips, allowing for corrections due to difference in tip height. Also, the potential ability to detect the slope of the surface seems very interesting.

For these reasons I recommend accepting the paper for publication, provided the authors address the following:

1) The main goal of this technique is improved imaging speed, but the authors never show what speed they have reached, or what speed could potentially be reached. What is for instance the exposure time for one a single frame? What is the time for the entire raster, and what was the data processing time? How does these pixel rates and full image times compare to regular AFM?

We appreciate the comment and agree that a discussion of current and potential speed should be presented. We have included this as a new paragraph in the discussion:

A driving consideration for parallelization is throughput, thus imaging speed is a main concern. Here, each frame was acquired in 1 s, which indicates that the imaging bandwidth of this system (defined as the number of points in a frame divided by the time to acquire one frame) was over 1 kHz. To provide a frame of reference for this number, the imaging bandwidth of a cantilever-based system is bounded by its mechanical bandwidth, which is approximated by the cantilever resonance frequency divided by cantilever quality factor. For standard cantilever-based probes this can range from ~200 Hz to 1 kHz and is a fundamental limitation of the cantilever, although it is worth highlighting achievements in the field of high speed cantilever-based AFM that utilize extremely high operating frequencies and custom designed stages for fast motion⁷. In contrast, the bandwidth of cantilever-free systems can be increased through parallelization and shortening frame acquisition time. As previously discussed, scaling to arrays that are 100× larger can be achieved with no change to the optical system. Due to the simplicity of the optical measurement, we predict that the measurement scheme may be reduced to 100 ms or less without specialized optics. With these improvements, MHz bandwidth is attainable. We note that total imaging time will also be defined by the number of frames in an image and processing time for the software. For instance, the data in Figure 4c took 256 s to collect and 54 s to process using the image processing algorithm described in Figure 3.

2) I think the authors should be a bit more admitting to the drawbacks of this measurement compared to regular AFM. The two main issues I would like mentioned / discussed is the lack of force control for the individual tips and the increased risk of tip damage. Regarding the first point, if I understand it correctly a 100 nm high feature in the sample will necessarily be imaged with 1000 nN more force on its highest part compared to the lowest. What implications does this have on the usefulness of the measurement? Regarding the second point, given that there are 1000 tips instead of one, the risk of at least one breaking must be very high. How stable were the polymer tips? Did you see any degradation during imaging?

We agree that the drawbacks of the approach should be made apparent in the manuscript and, through the comments raised by this Reviewer and the other two Reviewers, we have now explicitly addressed issues in terms of sample flatness, vertical resolution, and probe deformation. In order to more

explicitly raise this issue and the potential for addressing this concern, we have added the following passage to the discussion,

Parallelization, while increasing imaging bandwidth, also enforces the main limitation of this approach in the inability of probes to move independently to modulate probe-sample force or accommodate very tall features. In probes arrays such as the ones discussed here, this will impose restrictions on sample flatness and vertical range. However, it is worth highlighting work showing that cantilever-free arrays need not be passive, as scanning probe lithography experiments have shown that such arrays can be independently modulated using light, heat, or pneumatic systems^{13,23,33}, potentially providing a path to overcoming the challenges imposed by parallelization.

Regarding probes breaking, in all our imaging experiments, we were able to recover consistent images from all 1088 probes, thus we did not observe any probe breakage – although we did observe micron-scale variations in probe height that were corrected by our image processing algorithm. The more specific issue of probe wear was addressed in our response to Reviewer 1. We note, however, that if a probe did break, it would be possible to enlarge the region scanned by all probes so that isolated missing probes are covered by their neighbors. While this would lengthen imaging times by virtue of needing more pixels for each probe, it allows an array to be salvaged if one probe – or even many probes – are missing. We have added a note addressing this point:

While we did not observe the breakage of any probes during imaging, we note that the ability of neighboring probes to overlap their imaging areas provides this approach a potential method for dealing with broken probes by increasing the imaging area such that nominally each region is imaged by multiple probes.

3) Minor point: I was not aware of the Hough transform prior to reading the paper, and I think many other potential readers will be in the same position. A brief description of how it works and what it actually does would be useful. Placed either where it is first mentioned (page 4) or in the Methods part.

We thank the Reviewer for the suggestion and have added descriptions in the two places where the technique is addressed. Specifically, in the text when the technique is introduced, we added a parenthetical,

... Hough transform (an image processing technique that identifies circular features)...

Subsequently, we added a lengthier explanation in the methods section when the technique is introduced,

The Hough transform is a linear transformation that is applied to a matrix to locate circular features. Here, the process entails specifying a radius of interest in units of pixels. Then, a transformed image is produced in which, the value of each pixel is determined by the intensities of the pixels in the original image on a circle with the specified radius centered on the pixel's position.

Reviewers' Comments:

Reviewer #1:

Remarks to the Author:

The revised manuscript is somewhat stronger. There are a few technical issues that remain before the paper can be accepted.

The analysis of crosstalk is improved. The crosstalk will increase when the surface features get larger. Thus the 29% (or 35%) number is accurate for one specific feature height. The crosstalk will be larger for larger feature heights, and the nonlinear deformation in the elastic backing indicates that the crosstalk will not be proportional to the feature height. The crosstalk gets larger with larger features. At some point, the crosstalk makes imaging impractical, and already 29% is too large for industrial inspection applications. I would like to see a plot of neighbor tip imaging error vs feature height.

There is a contact stiffness between the tip and the substrate, and this contact stiffness is a function of the tip shape and properties as well as the substrate properties. It may be very small compared to the other stiffnesses in the system, however this contact stiffness should be calculated and analyzed compared to the other stiffnesses in the system. The authors state "contact mechanics indicates that the cantilever-free probe spring constant $0EF$ is only dependent upon the probe base radius R , and backing layer effective elastic modulus E ." This statement is not completely accurate and a bit misleading. This should be clarified before publication.

Reviewer #2:

Remarks to the Author:

The author has provided detail responses to the comments and revised the manuscript nicely. I feel that the manuscript is ready for publication.

Reviewer #3:

Remarks to the Author:

All my concerns have been addressed to my satisfaction. I recommend the paper for publication.

REVIEWER COMMENTS

Reviewer #1:

The revised manuscript is somewhat stronger. There are a few technical issues that remain before the paper can be accepted.

The analysis of crosstalk is improved. The crosstalk will increase when the surface features get larger. Thus the 29% (or 35%) number is accurate for one specific feature height. The crosstalk will be larger for larger feature heights, and the nonlinear deformation in the elastic backing indicates that the crosstalk will not be proportional to the feature height. The crosstalk gets larger with larger features. At some point, the crosstalk makes imaging impractical, and already 29% is too large for industrial inspection applications. I would like to see a plot of neighbor tip imaging error vs feature height.

We thank the Reviewer for raising this issue as it is an important item to make clear in the manuscript. Critically, in the limit of modest deformation relative to the probe heights (*i.e.* the imaging regime as described in the text), crosstalk will not change with surface feature height. This assertion is based fundamentally on the linearity of continuum mechanics in the small deformation limit and supported by a series of new simulations that we performed in which we varied the deformation of the center probe and observed a consistent fractional crosstalk in neighboring probes. The predictability of this crosstalk makes it facile to remove using the desharpening process described in the text. To make this clear, we have added an inset panel to Figure S6 showing the results of these simulations and added the caption:

Right inset shows the results of repeated simulations with different center probe deformations δ_z , showing the linearity of this relationship.

In addition, we have added the following passage in the main text to make the linearity of this relationship clear.

These simulations confirm a critical feature of crosstalk, specifically that probes move a consistent fraction of the distance of their neighbors. This linearity is key as it allows imaging artifacts from crosstalk to be removed using linear image processing, thus enabling the production of an image free from crosstalk artifacts.

There is a contact stiffness between the tip and the substrate, and this contact stiffness is a function of the tip shape and properties as well as the substrate properties. It may be very small compared to the other stiffnesses in the system, however this contact stiffness should be calculated and analyzed compared to the other stiffnesses in the system. The authors state "contact mechanics indicates that the cantilever-free probe spring constant $0EF$ is only dependent upon the probe base radius R , and backing layer effective elastic modulus E ." This statement is not completely accurate and a bit misleading. This should be clarified before publication.

We agree with the Reviewer that the probe-substrate contact will exhibit a contact stiffness. Indeed, this is explicitly included in the finite element analysis (FEA) that we performed to study the mechanics of imaging with cantilever-free probes (Figure S5). This analysis will allow us to add more details to the statement referenced by the Reviewer to provide the necessary caveats. Specifically, for our system, provided the sample has a modulus that is 1 GPa or stiffer, the backing layer deformation is an order of

magnitude larger than other deformations in the system, showing that the backing layer stiffness is dominating the contact mechanics. That being said, when atomic force microscope probes are described in a general sense, they are characterized by a probe stiffness that is agnostic to the substrate. Such descriptions are typically made with the understanding that the sample and probe tip may experience some deformation. This is because the spring constant is what allows one to relate an observed deformation to a force. Whether or not the probe is deforming, this is true in our case as well as the spring constant defined by the backing layer will relate the probe-sample force to the motion of the backing layer. We have added the following passage after the noted statement to acknowledge the Reviewer's point and make our assumptions clear:

It should be noted that while this relationship is expected to accurately relate the deformation of the backing layer to the probe-sample force, this does not preclude the probe or surface from deforming during an imaging experiment, an occurrence which is common to all forms of AFM. Such deformation effectively softens the probe-sample spring constant (*i.e.* the proportionality between the vertical motion of the z-piezo and the probe-sample force). For the system considered here, simulation predicts that this softening will be less than 10% provided the substrate stiffness is greater than 1 GPa (Figure S5).

Reviewer #2:

The author has provided detail responses to the comments and revised the manuscript nicely. I feel that the manuscript is ready for publication.

Reviewer #3:

All my concerns have been addressed to my satisfaction. I recommend the paper for publication.

Upon final review, we noted that an absolute value sign was missing from Equation S8, which we have now added.

Reviewers' Comments:

Reviewer #1:

Remarks to the Author:

The paper can be published in its current form.